

# *In vitro* and *ex vitro* propagation of Turkish myrtles through conventional and plantform bioreactor systems

Özhan Şimşek[1], Dicle Dönmez[2], Mehmet Ali Sarıdaş[3], Emine Acar[4], Yıldız Aka Kaçar[3], Sevgi Paydaş Kargı[3] and Tolga İzgü[5]

[1] Department of Horticulture, Erciyes University, Kayseri, Türkiye
[2] Biotechnology Research and Application Center, University of Çukurova, Adana, Türkiye
[3] Horticulture Department, University of Çukurova, Adana, Türkiye
[4] Biotechnology Department, Institute of Applied and Natural Sciences, University of Çukurova, Adana, Türkiye
[5] National Research Council of Italy (CNR), IBE/Institute of BioEconomy, Florence, Italy

## ABSTRACT

The myrtle (*Myrtus communis*) plant naturally grows in the temperate Mediterranean and subtropical regions and is used for various purposes; thus, it is among the promising species of horticultural crops. This study aimed to evaluate and compare the performance of different propagation systems, including rooting, solid media propagation, rooting, and with the Plantform bioreactor system, in achieving healthy and rapid growth of four myrtle genotypes with diverse genetic origins and well-regional adaptation. The selection of myrtle genotypes with distinct genetic backgrounds and proven adaptability to specific regions allowed for a comprehensive assessment of the propagation systems under investigation. Present findings proved that the Plantform system, the new-generation tissue culture system, was quite successful in micropropagation and rooting myrtle genotypes. We succeeded *in vitro* micropropagation and rooting of diverse wild myrtle genotypes, enabling year-round propagation without reliance on specific seasons or environmental conditions. The process involved initiating cultures from explants and multiplying them through shoot proliferation in a controlled environment. This contributes to sustainable plant propagation, preserving and utilizing genetic resources for conservation and agriculture.

# INTRODUCTION

The myrtle (*Myrtus communis*) plant, naturally growing in temperate Mediterranean and subtropical regions, belongs to the Myrtaceae family with 100 genera and 3000 species (*Rezaee & Kamali, 2014*). Myrtle essential oils are used in pharmaceuticals, and leaves are consumed as herbal tea (*Flamini et al., 2004*; *Dönmez, 2022*; *Şimşek et al., 2022*). Myrtle is a perennial evergreen shrub belonging to the Myrtaceae family, which exhibits a wide distribution in the Mediterranean region, often found in spontaneous bush-cover formations. Within the Myrtle species, two distinct subspecies are recognized: communis and tarentina. Notably, these subspecies can further be categorized into varieties based on

Corresponding author
Özhan Şimşek, ozhan12@gmail.com

the color of their ripe fruit, with melanocarpa characterized by a bluish-black peel and leucocarpa distinguished by a yellowish peel (*Medda & Mulas, 2021*). These species are monogenic, having hermaphrodite flowers. Myrtle leaves have an aromatic scent. Myrtle has berry-type fruits, mostly blackish-purple or white, which ripen in Autumn (October–December). Ripened fruits have a sugary and astringent taste and are pollinated by insects (*Şimşek et al., 2020*). The myrtle plant is used for various purposes; thus, it is among the promising species of horticultural crops. Leaf and fruit essential oils are commonly used in the chemical and pharmaceutical industries. Myrtle is also used for afforestation purposes on burned forests of the Mediterranean coastal zone and sold as an ornamental plant in European markets. Fruits are pretty popular in local bazaars of Türkiye. Because of these characteristics, fast, clonal, economic propagation-rooting methods should be developed to obtain healthy myrtle plants.

Although myrtle plants could be propagated from cuttings, seasonally different rooting levels are encountered. *In vitro* propagation is then used to overcome this problem. Bacterial and fungal contaminations are limited using plants developed *in vitro* tissue culture as an initial material.

Solid tissue culture increases the cost of propagated plants, and this method also has some adverse effects on the propagation coefficient and plant quality. Nutrient absorption from the shoots' basal sections is the most crucial disadvantage in nutrient uptake from the nutrient media. Therefore, it is asserted that liquid culture systems could constitute an effective alternative to solid or semi-solid media. However, the continuous placement of explants in liquid media brings various problems, such as tissue vitrification and asphyxiation. Contamination in plant tissue culture liquid systems poses a significant challenge in achieving successful plant propagation. Preventive measures such as strict aseptic conditions, proper sterilization techniques, and quality control are crucial. However, continuous vigilance and improved practices are necessary to minimize contamination and ensure successful plant tissue culture production in liquid systems (*Lambardi et al., 2015*).

A temporary Immersion System (TIS) combines the advantages of conventional semi-solid and liquid media. As compared to classical tissue culture systems, benefits of this system include more uniform contact of culture media and plant with nutrient media, less incidence of vitrification and asphyxiation, less presence of toxic compounds secreted by cultures and resulted in browning than the liquid media, periodical change of atmosphere in culture plates, prevention of accumulation of $CO_2$ and ethylene-like harmful gases, longer sub-culture durations since larger culture plates are used, the easy and prompt performance of culture as compared to solid nutrient media requiring greater attention, care, and labor, promotion of cell division with the aid of bubbles generated through air circulation (*Lambardi et al., 2015*). All these advantages then increase both propagation coefficient and shoot quality.

The Plantform temporary immersion system, also known as the Plantform bioreactor system, is a specialized plant tissue culture technique used for the large-scale propagation of plants *in vitro*. It is designed to automate the process of nutrient delivery to plant tissues and optimize their growth and development. The system comprises a series of culture vessels or containers connected to a central control unit. Each culture vessel typically

contains multiple plantlets or explants submerged in a liquid growth medium. The culture vessels are placed on a rotating platform that periodically immerses the plant material in the nutrient solution and drains it, mimicking the natural process of plant nutrient uptake. The key features of the Plantform temporary immersion system include: (1) Automated immersion and drainage: The system controls the timing and duration of immersion cycles, allowing the plant material to receive precise and optimal nutrient exposure. This enhances nutrient uptake, promotes growth, and improves the overall efficiency of the propagation process. (2) Sterility and aseptic conditions: The system is designed to maintain sterile and aseptic conditions to prevent contamination and ensure the health and integrity of the plant material. The culture vessels are sealed and equipped with filters to prevent the entry of airborne contaminants. (3) Nutrient optimization: The system allows for precise control over the composition and concentration of the nutrient medium. This enables researchers to tailor the medium to meet the specific requirements of the plant species and optimize growth and development. (4) Scalability: The Plantform system is designed for large-scale production, allowing for the simultaneous propagation of multiple plantlets or explants in separate culture vessels. This makes it suitable for commercial plant propagation and research applications requiring a high volume of plant material.

The Plantform temporary immersion system offers several advantages over traditional plant tissue culture techniques. It improves the efficiency of nutrient delivery, reduces labor and material costs, increases production capacity, and provides better control over the growth and development of plant material (*Georgiev et al., 2014*; *Welander et al., 2017*).

The "Plantform" is a temporary immersion bioreactor system, and differences from the other systems include easy-to-use, autoclavable, transparent, easy-to-carry, existence of gas exchange, air pump, and timers. Such differences improve the quality of the system of *in vitro* cultures. The system relies on air circulation within cultural places and a temporary immersion method. Immersion duration and frequency could be adjusted, and gas exchange could be done with a timer.

The first studies on Temporary Immersion Bioreactor Systems were reported 30 years ago, and different prototypes have been developed since then. For instance, diverse Temporary Immersion Reactors are used for clonal propagation in tissue culture, such as RITA (Recipient for Automated Temporary Immersion System) and BIT (Twin Flasks). The RITA system was first developed by *Alvard, Cote & Teisson (1993)*, *Pavlov & Bley (2006)*, and *Zhu, Li & Welander (2015)* performed some modifications. The BIC systems were developed by *Escalona et al. (1999)* and used for different plant species (*Escalona et al., 2003*; *Welander, Zhu & Li, 2007*). TIS bioreactors were employed in the mass propagation of strawberries (*Takayama & Akita, 1998*), ornamental plants (*Dewir et al., 2006*), Vaccinium angustifolium (*Debnath, 2009*), potatoes (*Piao et al., 2003*), and several other plant species. However, the bioreactors developed during this period were either exceedingly small or too heavy; thus, they needed to be more practical systems (*Welander, Zhu & Li, 2007*). Therefore, a new temporary immersion system (Plantform) has been developed in recent years to overcome the problems encountered in previous temporary immersion bioreactor systems.

**Table 1** Plant material information.

| Genotype name | Sampling location province/city/country | Fruit color |
|---|---|---|
| Erdemli Beyazı | Tömük/Erdemli/Mersin/Türkiye | White-fruited |
| Erdemli Siyahı | Erdemli/Mersin/Türkiye | Dark-fruited |
| Karaisalı Beyazı | Karaisalı/Adana/Türkiye | White-fruited |
| Karaisalı Siyahı | Karaisalı/Adana/Türkiye | Dark-fruited |

Plant domestication continues to be a valuable approach to enhancing multipurpose and sustainable agriculture. Researchers worldwide are actively studying the diverse array of wild plant species, specifically focusing on their potential for domestication. The process of domestication, driven by both natural selection and human intervention, along with environmental factors, induces notable alterations in plants' various morphological, phenotypic, and vegetative traits. These domesticated cultivars offer distinct advantages in improved agricultural productivity, enhanced adaptability, and desirable characteristics that cater to human needs. Consequently, understanding the mechanisms underlying domestication and the resultant changes in plant traits is pivotal for advancing crop improvement strategies, breeding programs, and overall agricultural sustainability. In addition to traditional classical propagation methods, *in vitro* propagation techniques have gained significant attention in recent years, offering a controlled environment for the rapid and efficient propagation of domesticated plant varieties (*Medda & Mulas, 2021*).

This article aims to contribute to the existing knowledge on plant domestication and provide insights for further research in this crucial field. In the present study, four myrtle genotypes with different genetic origins and well-regional adaptation were selected, and the performance of o rooting systems (rooting, solid media rooting, and rooting with the Planform bioreactor system) was compared to achieve a healthy and speedy rooting of myrtle plants.

## MATERIALS & METHODS

### Plant material

Visual assessments such as fruit color, size, and tree crown structure were made in selection studies conducted in the Adana and Mersin provinces of Türkiye. Four wild myrtle genotypes with high yield levels, large fruits, and healthy and robust plant structures were selected as the plant material of the present study. Two genotypes are black, and the other has white fruits (Table 1). The authors complied with the IUCN Policy Statement on Research Involving Species at Risk of Extinction and the Convention on the Trade in Endangered Species of Wild Fauna and Flora for collecting the plant material. The plant material was collected with the official permission numbered 43368836-335.01-E.1189927 obtained from the Ministry of Agriculture and Forestry, the Republic of Türkiye.

### Tissue culture works
### Sterilization of shoot tips

Shoot tips were sterilized before taking them into a culture. Shoot tips were initially washed through tap water for 1 min. Then, shoot tips were immersed in 70% ethyl alcohol for 3 min, then in 20% sodium hypochlorite solution for 1 min. The shoot tips were washed through sterile distilled water 3 times in a sterile cabin to remove sterilant materials.

### Plant culture and micropropagation

After sterilization, shoot tips were cultured in MS nutrient medium containing 0, 1, 2 mg/L BAP, and 8 g/L agar. Plants were cultured under $25 \pm 2$ °C temperature and 16/8 h (light/dark) photoperiods. Plants were taken into sub-cultures every four weeks. Three sub-cultures were practiced throughout the experiments.

### Solid culture rooting

Solid culture rooting trails were conducted in MS nutrient media containing 0, 1, 2 mg/L IBA and 8 g/L agar. Plants were cultured under $25 \pm 2$°C temperature and 16/8 h (light/dark) photoperiods. Plants were cultured in solid media for six weeks (*Biçen, 2017*).

### Micropropagation and rooting in the plantform bioreactor system

For rooting trials, 0, 1, 2 mg/L BAP-containing MS media were used for micropropagation, and 0, 1, 2 mg/L IBA-containing MS media. Agar was not supplemented into nutrient media used in micropropagation and rooting trials of the Plantform bioreactor system. Each culture plate was supplemented with 500 ml nutrient medium. Culture plates were subjected to 10 min immersion every 8 h and 15 min aeration every 4 h. For micropropagation of myrtle plants in the Plantform culture plates, 3 sub-cultures were practiced. For rooting, plants were cultured for seven weeks. Plants were cultured under $25 \pm 2$ °C temperature and 16/8 h (light/dark) photoperiods.

### Acclimatization of plants to external conditions

Before taking rooted plants into external conditions, lids of culture plates in the bioreactor system were gradually opened for pre-acclimatization in the laboratory. Plants were transplanted into vials containing a 1:1 sterile peat: perlite mixture in the greenhouse. Plants were acclimatized gradually to external conditions under a small mini tunnel. Plantlets coming from both systems were monitored in separate vials.

### Rooting

For rooting, 15 cm long semi-woody scions were used. Cuttings were taken in late Spring before the flowering and autumn seasons to put forth the season effects. These cuttings were immersed into Hormoril T-8 powder containing 0.8% IBA and 5% thiabendazole, and rooting trials were conducted in perlite-containing boxes (*Klein, Cohen & Hebbe, 2000*).

### Experimental design, statistical analyses, and investigated characteristics

Tissue culture trials were conducted in a $4 \times 4 \times 3$ factorial experimental design with three replications. For each myrtle genotype in the solid culture and the Plantform system, 90

shoot tip immersions were performed (30 shoot tips for each replicate). For scion rooting, 10 cuttings were used in each replicate of each genotype.

For micropropagation trials of myrtle plants in solid media and the Plantform system, multiplication coefficient, plant height (cm), and the number of leaves per plant features were measured. In rooting trials, rooting ratio (%), plant length (cm), the number of roots, and root lengths were measured eight weeks after culture initiation in rooting trials and 12 weeks after culture initiation in rooting trials.

Experimental data were subjected to analysis of variance with the use of JMP 5.01. The percent (%) values were subjected to arcsin transformation. Significant means were compared with the help of the LSD test. We applied multiple comparison tests at a 5% significance level.

# RESULTS
## Results of tissue culture
## Micropropagation

Shoot tips of four different myrtle genotypes (selected from Karaisalı/Adana/Türkiye and Erdemli/Mersin/Türkiye and with white and black fruits) were cultured in 0, 1 and 2 mg/L BAP-containing solid media and the Plantform system. The multiplication coefficient and plant heights (cm) of resulting plantlets were measured, and results are provided in Tables 2 and 3.

Regarding the multiplication coefficient of myrtle genotypes in solid media and Plantform, the best result was achieved from the Erdemli Beyazı genotype (7.35). Within this genotype, the Plantform system (11.66) yielded about four times greater multiplication coefficient than solid media (3.04). Regarding Genotype × BAP × Culture Plate interactions, the most significant multiplication coefficient (21.20) was obtained from 2 mg/L BAP-containing the Plantform system of the Karaisalı Beyazı genotype. In terms of multiplication coefficients at different BAP concentrations, the best outcomes were achieved from 2 mg/L BAP (9.83) and 1 mg/L BAP (8.28) concentrations. The media without BAP yielded a multiplication coefficient of only 1.20.

In terms of plant heights of micropropagated myrtle genotypes, the best outcomes were achieved from Erdemli Beyazı (3.92 cm), Karaisalı Beyazı (3.50 cm), and Karaisalı Siyahı (4.09 cm), which were placed into the same statistical group. Erdemli Siyahı (2.88 cm) yielded relatively shorter explants than the others. The Plantform system (5.80 cm) yielded about four times greater plant heights than the solid culture (1.40 cm). As a multiplication coefficient, the Plantform system offered a significant advantage in plant height. Regarding the effects of BAP concentrations on plant height, it was observed that the media without BAP yielded successful outcomes (4.37 cm). The average plant height was 3.49 cm for 1 mg/L BAP-containing media and 2.93 cm for 2 mg/L BAP-containing media.

### *In vitro* rooting

Plantlets were cultured in 0, 1, and 2 mg/L IBA-containing solid culture and the Plantform system. Plantlets obtained through micropropagation of myrtle genotypes in solid culture

**Table 2  Multiplication coefficients of myrtle genotypes micro-propagated in solid media and Plantform system.**

| Genotype | Culture plate | BAP concentration | G*CP*BAP | G*CP | G |
|---|---|---|---|---|---|
| Erdemli Beyazı | Plantform | BAP 0 mg/L | 1.60 | | |
| | | BAP 1 mg/L | 16.40 | 11.66a | |
| | | BAP 2 mg/L | 17.00 | | 7.35 |
| | Solid Culture | BAP 0 mg/L | 1.00 | | |
| | | BAP 1 mg/L | 2.13 | 3.04bc | |
| | | BAP 2 mg/L | 6.00 | | |
| Erdemli Siyahı | Plantform | BAP 0 mg/L | 1.00 | | |
| | | BAP 1 mg/L | 9.60 | 5.86b | |
| | | BAP 2 mg/L | 7.00 | | 4.84 |
| | Solid Culture | BAP 0 mg/L | 1.00 | | |
| | | BAP 1 mg/L | 5.26 | 3.82bc | |
| | | BAP 2 mg/L | 5.19 | | |
| Karaisalı Beyazı | Plantform | BAP 0 mg/L | 1.40 | | |
| | | BAP 1 mg/L | 11.80 | 11.46a | |
| | | BAP 2 mg/L | 21.20 | | 6.60 |
| | Solid Culture | BAP 0 mg/L | 1.00 | | |
| | | BAP 1 mg/L | 1.00 | 1.73c | |
| | | BAP 2 mg/L | 3.20 | | |
| Karaisalı Siyahı | Plantform | BAP 0 mg/L | 1.60 | | |
| | | BAP 1 mg/L | 16.00 | 10.73a | |
| | | BAP 2 mg/L | 14.60 | | 6.96 |
| | Solid Culture | BAP 0 mg/L | 1.00 | | |
| | | BAP 1 mg/L | 4.10 | 3.20bc | |
| | | BAP 2 mg/L | 4.50 | | |

**Notes.**

$LSD_{CP}$:1.70*** $LSD_{BAP}$:2.09*** $LSD_G$: N.S. $LSD_{G*CP*BAP}$: N.S. $LSD_{G*CP}$:1.54** $LSD_{CP*BAP}$: 2.95*** $LSD_{G*BAP}$: N.S.

**$p < 0.01$.

***$p < 0.001$.

N.S., Not Significant.

and the Plantform system were included in these rooting trials. Eight weeks after the initiation of rooting trials, rooting ratio (%), the number of roots, root length (cm), and plant heights (cm) were measured. Results are provided in Tables 4, 5, 6 and 7. Regarding *in vitro* rooting trials of the myrtle genotypes, Erdemli Beyazı yielded the best rooting ratio (60.70%).

Regarding *in vitro* rooting trials of the myrtle genotypes, Erdemli Beyazı yielded the best rooting ratio (60.70%). Regarding the Rooting percentage of culture plates, solid culture yielded a rooting ratio of 58.00%, and the Plantform system yielded a rooting ratio of 50.00%. The most significant value (80.62%) for the Rooting percentage of IBA concentrations was obtained from 2 mg/L IBA concentration. It was followed by 1 mg/L IBA (79.92%) concentration, and the lowest value (2.00%) was obtained from the media without IBA.

**Table 3  Plant heights (cm) of myrtle genotypes micropropagated in solid media and Plantform system.**

| Genotype | Culture plate | BAP concentration | G*CP*BAP | G*CP | G |
|---|---|---|---|---|---|
| Erdemli Beyazı | Plantform | BAP 0 mg/L | 10.60a | | |
| | | BAP 1 mg/L | 4.60cd | 6.46a | |
| | | BAP 2 mg/L | 4.20de | | |
| | Solid Culture | BAP 0 mg/L | 0.22ı | | 3.92a |
| | | BAP 1 mg/L | 1.94fgh | 1.37c | |
| | | BAP 2 mg/L | 1.96fgh | | |
| Erdemli Siyahı | Plantform | BAP 0 mg/L | 4.32de | | |
| | | BAP 1 mg/L | 5.20cd | 4.17b | |
| | | BAP 2 mg/L | 3.00ef | | |
| | Solid Culture | BAP 0 mg/L | 1.28ghı | | 2.88b |
| | | BAP 1 mg/L | 2.14fg | 1.60c | |
| | | BAP 2 mg/L | 1.39ghı | | |
| Karaisalı Beyazı | Plantform | BAP 0 mg/L | 8.30b | | |
| | | BAP 1 mg/L | 5.50cd | 6.10a | |
| | | BAP 2 mg/L | 4.50d | | |
| | Solid Culture | BAP 0 mg/L | 0.50 hı | | 3.50a |
| | | BAP 1 mg/L | 0.90 ghı | 0.90c | |
| | | BAP 2 mg/L | 1.30ghı | | |
| Karaisalı Siyahı | Plantform | BAP 0 mg/L | 8.20b | | |
| | | BAP 1 mg/L | 6.00c | 6.45a | |
| | | BAP 2 mg/L | 5.20cd | | |
| | Solid Culture | BAP 0 mg/L | 1.58fghı | | 4.09a |
| | | BAP 1 mg/L | 1.70fgh | 1.72c | |
| | | BAP 2 mg/L | 1.90fgh | | |

**Notes.**

$LSD_{CP}$: 0.42*** $LSD_{BAP}$: 0.52*** $LSD_G$: 0.60*** $LSD_{G*CP*BAP}$: 1.47*** $LSD_{G*CP}$: 0.85*** $LSD_{CP*BAP}$: 0.73*** $LSD_{G*BAP}$: 1.04***.

\*\*\* $p < 0.001$.

Differences in the number of roots per plant of myrtle genotypes in the *in vitro* rooting trials were not found to be significant. However, the number of roots in the Plantform system and solid culture differed significantly. While plantlets generated an average of 3.20 roots in the Plantform system, the average number of roots developed in solid culture was 1.95. Regarding IBA concentrations, the most considerable number of roots (3.77) was obtained from 1 mg/L IBA concentration.

Regarding root lengths, the Karaisalı Siyahı (3.96 cm) genotype was identified as the most successful. The Plantform system yielded better root lengths than the solid culture. The average root length was identified as 4.29 cm for the Plantform system and 2.07 cm for the solid culture. Regarding IBA concentrations, the average root length was determined as 4.41 cm for 1 mg/IBA concentration and 4.51 cm for 2 mg/IBA concentration.

Regarding plant heights in the *in vitro* rooting trials, Karaisalı Beyazı (4.69 cm) was identified as the most successful genotype. The average plant height was 4.43 cm for the Plantform system and 3.42 cm for the solid culture. In terms of IBA concentrations, the

Table 4 Rooting ratios (%) of myrtle genotypes cultured in solid media and Plantform system.

| Genotype | Culture plate | BAP concentration | G*CP*BAP | G*CP | G |
|---|---|---|---|---|---|
| Erdemli Beyazı | Plantform | IBA 0 mg/L | 6.00 (14.00[a])₁ | 61.33a (54.64) | 60.70a (52.42) |
| | | IBA 1 mg/L | 98.00ab (86.31) | | |
| | | IBA 2 mg/L | 80.00ef (63.61) | | |
| | Solid Culture | IBA 0 mg/L | 5.20₁(11.69) | 60.06a (50.19) | |
| | | IBA 1 mg/L | 88.00de (69.82) | | |
| | | IBA 2 mg/L | 87.00de (69.07) | | |
| Erdemli Siyahı | Plantform | IBA 0 mg/L | 0.00j (0.00) | 51.66b (41.37) | 52.56b (46.72) |
| | | IBA 1 mg/L | 84.00de (66.69) | | |
| | | IBA 2 mg/L | 71.00fg (57.43) | | |
| | Solid Culture | IBA 0 mg/L | 0.00 (0.00) | 53.46a (52.07) | |
| | | IBA 1 mg/L | 60.40de (66.21) | | |
| | | IBA 2 mg/L | 100.00a (90.00) | | |
| Karaisalı Beyazı | Plantform | IBA 0 mg/L | 0.00 (0.00) | 43.33 (35.87)c | 52.50b (43.87) |
| | | IBA 1 mg/L | 67.00fgh (55.02) | | |
| | | IBA 2 mg/L | 63.00gh (52.59) | | |
| | Solid Culture | IBA 0 mg/L | 1.00j (2.58) | 61.66a (51.87) | |
| | | IBA 1 mg/L | 88.00cde (71.88) | | |
| | | IBA 2 mg/L | 97.00ab (83.73) | | |
| Karaisalı Siyahı | Plantform | IBA 0 mg/L | 10.00₁(18.44) | 44.00bc (40.45) | 52.00b (46.11) |
| | | IBA 1 mg/L | 66.00gh (54.34) | | |
| | | IBA 2 mg/L | 56.00 h (48.57) | | |
| | Solid Culture | IBA 0 mg/L | 0.00j (0.00) | 60.00a (51.77) | |
| | | IBA 1 mg/L | 88.00cd (74.06) | | |
| | | IBA 2 mg/L | 91.00bc (78.66) | | |

**Notes.**

$LSD_{CP}$:2.49*** $LSD_{IBA}$:3.05*** $LSD_{G}$:3.53*** $LSD_{G*CP*IBA}$:8.65*** $LSD_{G*CP}$:4.99*** $LSD_{CP*IBA}$:4.32*** $LSD_{G*IBA}$:6.11***.

*** $p < 0.001$.

[a]Percentalies were subjected to angle transformation and these values were presented in parentheses.

media without IBA yielded better outcomes. Myrtle plants rooted in Plantform systems and solid culture were acclimatized to outside conditions. Plants were initially taken into mini greenhouses. After about 10 days, plants were removed from these greenhouses and put under the misting unit. All the rooted plants from solid culture and the Plantform system were successfully acclimatized to outer conditions. After about a month, acclimatized myrtle plants were transplanted into the pots. Some figures of various stages of myrtle micropropagation are presented in Fig. 1.

## Rooting

The rooting potentials of the cuttings taken from the selected myrtle genotypes were investigated in the present study. The cuttings of four different myrtle genotypes were taken in two different periods, and rooting trials were conducted in two different periods accordingly. Semi-woody cuttings one cm long were used in rooting trials. To put forth the effect of the period, cuttings were taken in late Spring before flowering (May–June)

**Table 5  Number of roots per plant of myrtle genotypes cultured in solid media and Plantform system.**

| Genotype | Culture plate | BAP concentration | G*CP*BAP | G*CP | G |
|---|---|---|---|---|---|
| Erdemli Beyazı | Plantform | IBA 0 mg/L | 0.40 | | |
| | | IBA 1 mg/L | 4.20 | 2.53bcd | |
| | | IBA 2 mg/L | 3.00 | | 2.50 |
| | Solid Culture | IBA 0 mg/L | 1.00 | | |
| | | IBA 1 mg/L | 3.40 | 2.46bcde | |
| | | IBA 2 mg/L | 3.00 | | |
| Erdemli Siyahı | Plantform | IBA 0 mg/L | 0.80 | | |
| | | IBA 1 mg/L | 4.00 | 2.86bc | |
| | | IBA 2 mg/L | 3.80 | | 2.43 |
| | Solid Culture | IBA 0 mg/L | 0.00 | | |
| | | IBA 1 mg/L | 3.20 | 2.00cde | |
| | | IBA 2 mg/L | 2.80 | | |
| Karaisalı Beyazı | Plantform | IBA 0 mg/L | 0.00 | | |
| | | IBA 1 mg/L | 4.80 | 3.20b | |
| | | IBA 2 mg/L | 4.79 | | 2.36 |
| | Solid Culture | IBA 0 mg/L | 0.00 | | |
| | | IBA 1 mg/L | 3.00 | 1.53e | |
| | | IBA 2 mg/L | 1.60 | | |
| Karaisalı Siyahı | Plantform | IBA 0 mg/L | 1.60 | | |
| | | IBA 1 mg/L | 4.40 | 4.20a | |
| | | IBA 2 mg/L | 6.60 | | 3.00 |
| | Solid Culture | IBA 0 mg/L | 0.20 | | |
| | | IBA 1 mg/L | 3.20 | 1.80de | |
| | | IBA 2 mg/L | 2.00 | | |

**Notes.**
$LSD_{CP}$: 0.46*** $LSD_{IBA}$:0.57*** $LSD_{G}$:N.S. $LSD_{G*CP*IBA}$:N.S. $LSD_{G*CP}$:0.93** $LSD_{CP*IBA}$:0.81** $LSD_{G*IBA}$:N.S.

** $p < 0.01$.

*** $p < 0.001$.

N.S, Not Significant.

and in Autumn within the fruit ripening period (October–November). The rooting of myrtle cuttings is presented in Fig. 2. Rooting was not encountered in all genotypes with the cuttings in Autumn. Then, trials were set up again with Spring cuttings, and results are provided in Table 8.

In terms of rooting results of myrtle genotypes for cuttings taken in the Spring season, rooting was not encountered in Erdemli Siyahı and Karaisalı Beyazı genotypes, Karaisalı Siyahı had a rooting ratio of 13.33%, and Erdemli Beyazı had a rooting ratio of 3.33%. Rooting results achieved in solid culture and the Plantform system under *in vitro* conditions were compared with the rooting results, and resultant data are provided in Table 9.

The differences between the *in vitro* and rooting results of the genotypes were found to be significant. In this sense, the Erdemli Beyazı genotype (71.26%) yielded a significantly greater rooting ratio than the other genotypes. Besides, cuttings taken in May produced a

**Table 6  Root lengths (cm) of myrtle genotypes cultured in solid media and Plantform system.**

| Genotype | Culture plate | BAP concentration | G*CP*BAP | G*CP | G |
|---|---|---|---|---|---|
| Erdemli Beyazı | Plantform | IBA 0 mg/L | 0.40ıj | | |
| | | IBA 1 mg/L | 2.54fgh | 2.21c | |
| | | IBA 2 mg/L | 3.70ef | | 2.14c |
| | Solid Culture | IBA 0 mg/L | 1.00hıj | | |
| | | IBA 1 mg/L | 1.54hıj | 2.08c | |
| | | IBA 2 mg/L | 3.70ef | | |
| Erdemli Siyahı | Plantform | IBA 0 mg/L | 1.20hıj | | |
| | | IBA 1 mg/L | 7.40b | 4.46b | |
| | | IBA 2 mg/L | 4.80de | | 3.26b |
| | Solid Culture | IBA 0 mg/L | 0.00j | | |
| | | IBA 1 mg/L | 1.80ghı | 2.06c | |
| | | IBA 2 mg/L | 4.40de | | |
| Karaisalı Beyazı | Plantform | IBA 0 mg/L | 0.00j | | |
| | | IBA 1 mg/L | 7.40b | 4.33b | |
| | | IBA 2 mg/L | 5.60cd | | 3.35ab |
| | Solid Culture | IBA 0 mg/L | 0.00j | | |
| | | IBA 1 mg/L | 3.80ef | 2.36c | |
| | | IBA 2 mg/L | 3.30efg | | |
| Karaisalı Siyahı | Plantform | IBA 0 mg/L | 2.30fgh | | |
| | | IBA 1 mg/L | 6.60bc | 6.16a | |
| | | IBA 2 mg/L | 9.60a | | 3.96a |
| | Solid Culture | IBA 0 mg/L | 0.00j | | |
| | | IBA 1 mg/L | 4.20de | 1.76c | |
| | | IBA 2 mg/L | 1.00hıj | | |

**Notes.**
LSD$_{CP}$:0.44*** LSD$_{IBA}$: 0.54*** LSD$_G$: 0.63*** LSD$_{G*CP*IBA}$:1.55*** N.S. LSD$_{G*CP}$: 0.89*** LSD$_{CP*IBA}$: 0.39*** LSD$_{G*IBA}$:1.09***.
*** $p < 0.001$.

substantially lower rooting ratio (4.16%) than tissue culture Rooting percentage. Regarding Rooting percentage of tissue culture methods, the most excellent rooting ratio (93.75%) was obtained from 1 mg/L IBA-containing solid culture medium (solid culture 2).

# DISCUSSION

Modern biotechnological methods, including tissue culture techniques, are applied to achieve better results in plant breeding. Today, tissue culture techniques constitute practical tools for developing new cultivars through breeding (*Dönmez, Şimşek & Aka Kaçar, 2016*). Among the tissue culture techniques, micro-propagation has been increasingly used in speedy clonal propagation and rooting of various species.

Most micropropagation and rooting trials of tissue culture studies have been conducted with solid cultures. Full-liquid media have then started to be used in time because of some disadvantages of solid cultures. However, the continuous placement of explants in liquid media brings about various problems, such as tissue vitrification and asphyxiation

**Table 7  Plant height (cm) of myrtle genotypes cultured in solid media and Plantform system.**

| Genotype | Culture plate | BAP concentration | G*CP*BAP | G*CP | G |
|---|---|---|---|---|---|
| Erdemli Beyazı | Plantform | IBA 0 | 6.60ab | | |
| | | IBA 1 | 4.20defg | 4.83ab | |
| | | IBA 2 | 3.70efgh | | 4.18ab |
| | Solid Culture | IBA 0 | 5.50abcde | | |
| | | IBA 1 | 3.20fghı | 3.53c | |
| | | IBA 2 | 1.90hıj | | |
| Erdemli Siyahı | Plantform | IBA 0 | 4.76bcdef | | |
| | | IBA 1 | 2.20hıj | 3.18c | |
| | | IBA 2 | 2.60ghıj | | 3.32c |
| | Solid Culture | IBA 0 | 5.56abcde | | |
| | | IBA 1 | 2.20hıj | 3.45c | |
| | | IBA 2 | 2.60ghıj | | |
| Karaisalı Beyazı | Plantform | IBA 0 | 5.24abcde | | |
| | | IBA 1 | 5.40abcde | 5.88a | |
| | | IBA 2 | 7.00a | | 4.69a |
| | Solid Culture | IBA 0 | 4.34defg | | |
| | | IBA 1 | 3.66efgh | 3.50c | |
| | | IBA 2 | 2.50ghıj | | |
| Karaisalı Siyahı | Plantform | IBA 0 | 5.80abcd | | |
| | | IBA 1 | 1.30ıj | 3.83bc | |
| | | IBA 2 | 4.40cdefg | | 3.55bc |
| | Solid Culture | IBA 0 | 6.30abc | | |
| | | IBA 1 | 2.50ghıj | 3.26c | |
| | | IBA 2 | 1.00j | | |

**Notes.**
$LSD_{CP}$: 0.55*** $LSD_{IBA}$: 0.67*** $LSD_{G}$: 0.78** $LSD_{G*CP*IBA}$: N.S. $LSD_{G*CP}$:1.10** $LSD_{CP*IBA}$:0.95** $LSD_{G*IBA}$:1.35**.
** $p < 0.01$.
*** $p < 0.001$.
N.S., Not Significant.

**Table 8  Rooting ratio (%), number of roots per plant and root lengths (cm) of scions taken from myrtle plants.**

| Genotype | Rooting ratio (%) | Number of roots per plant | Root length (cm) |
|---|---|---|---|
| Erdemli Beyazı | 3.33 (6.14[a]) | 0.33 | 0.66b |
| Erdemli Siyahı | 0.00 (0.00) | 0.00b | 0.00b |
| Karaisalı Beyazı | 0.00 (0.00) | 0.00b | 0.00b |
| Karaisalı Siyahı | 13.33 (21.14) | 2.33a | 6.33a |

**Notes.**
$LSD_{RR}$: 10.95** $LSD_{NRP}$: N.S. $LSD_{RL}$: 1.21***.
** $p < 0.01$.
*** $p < 0.001$.
N.S., Not Significant.
[a] Percentalies were subjected to angle transformation and these values were presented in parentheses.
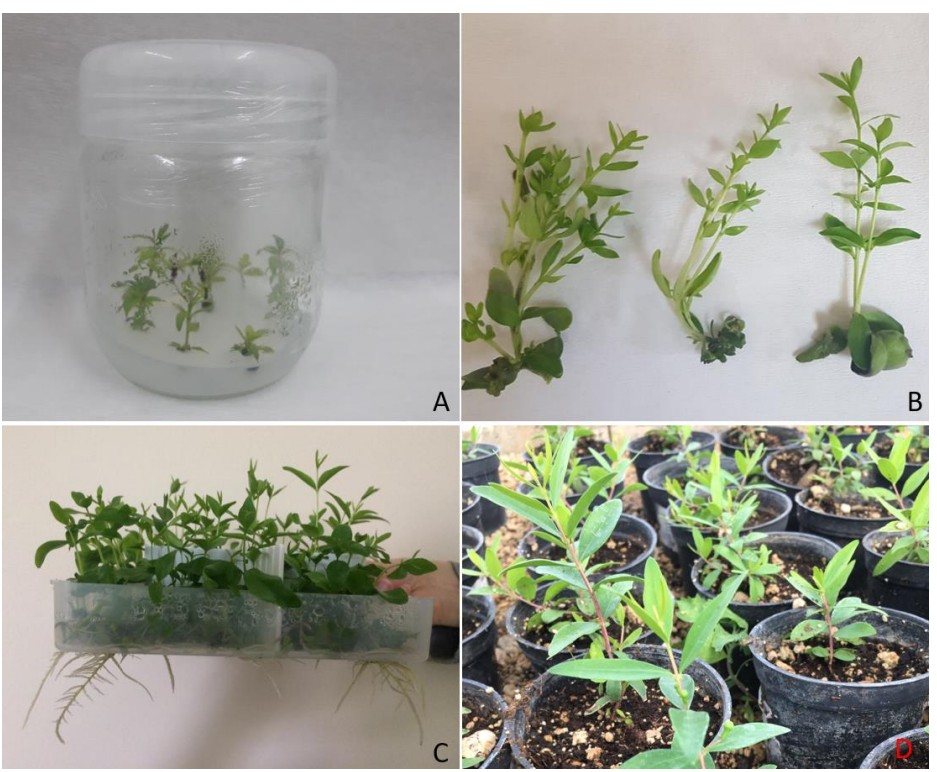

**Figure 1** (A) Myrtle plants in solid medium, (B) myrtle plants coming from Plantform, (C) rooting in Plantform, (D) acclimatized myrtle plants.

**Table 9** Comparison of rooting ratios achieved in solid culture and Plantform system under *in vitro* conditions with the rooting ratios achieved in scion rooting trials.

| Genotype Treatment | Erdemli Beyazı | Erdemli Siyahı | Karaisalı Beyazı | Karaisalı Siyahı | Treatment average |
|---|---|---|---|---|---|
| Cutting | 0.00k (0.00[a]) | 3.33j (6.14) | 0.00k (0.00) | 13.33ı(21.14) | 4.16d (6.82) |
| Plantform 1 IBA | 98.00b (81.87) | 84.00de (66.42) | 67.00fg (54.93) | 66.00fg (54.33) | 78.75b (64.39) |
| Plantform 2 IBA | 80.00e (63.43) | 71.00f (57.41) | 63.00gh (52.53) | 56.00 h (48.44) | 67.50c (55.45) |
| Solid Culture 1 IBA | 88.00cd (69.73) | 60.40gh (51.00) | 88.00cd (69.73) | 88.00cd (69.73) | 81.10b (65.05) |
| Solid Culture 2 IBA | 87.00cd (68.86) | 100.00a (90.00) | 97.00b (80.02) | 91.00c (72.54) | 93.75a (77.85) |
| Genotype Average | 71.26a (58.01) | 63.08b (52.96) | 63.00b (51.44) | 62.86b (53.23) | |

**Notes.**

$LSD_{Treatment}$: 2.14*** $LSD_G$:1.91*** $LSD_{Treatment*G}$: 2.26***.

*** $p < 0.001$.

[a]Percentalies were subjected to angle transformation and these values were presented in parentheses.

(*Lambardi et al., 2015*). Therefore, temporary immersion reactor systems offering a transition between solid and liquid media have been developed and commonly used in tissue culture studies.

The number of studies conducted using the Plantform temporary immersion bioreactor system in myrtle plants is minimal. Two previous studies indicated that the system yielded

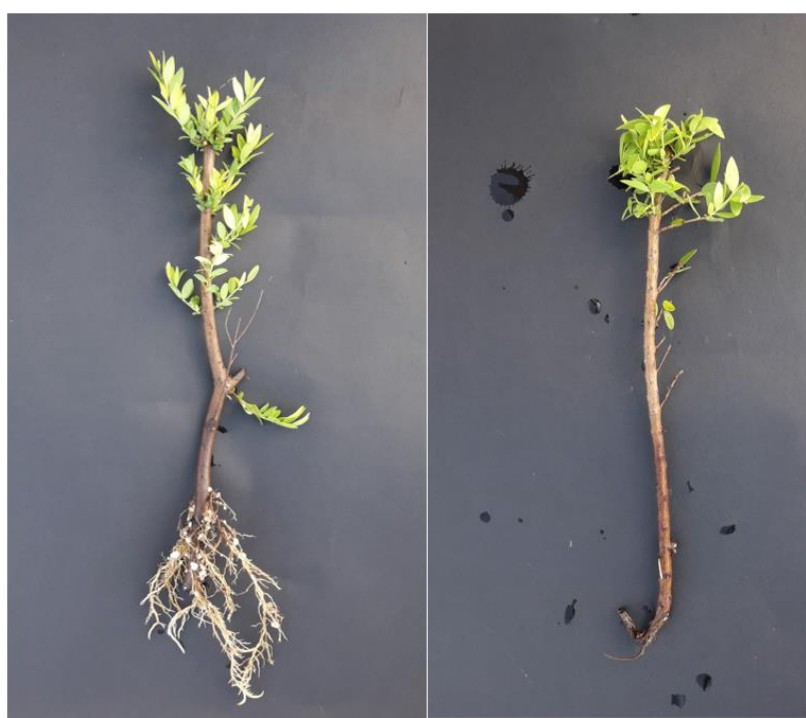

**Figure 2  Rooting of cuttings (left is well-rooted, right is not well-rooted cuttings).**

better outcomes in myrtle plants than semi-solid and/or solid media (*Aka Kaçar et al., 2020*; *Benelli, Fernanda & De Carlo, 2015*). In the present study, four different myrtle genotypes with varying fruit colors were micro-propagated and rooted in solid cultures and the Plantform temporary immersion bioreactor system. Rooting of the cuttings taken in two periods (Autumn and late Spring) under external conditions was also investigated. Despite some differences between the genotypes, the plants developed in the Plantform system were more successful than the other treatments. In general, the Plantform system yielded better propagation and rooting outcomes.

Genotype is among the most significant factors influencing success in tissue culture studies. There were some differences in the propagation and rooting parameters of the present genotypes. Previous tissue culture studies in myrtle plants also revealed different outcomes for myrtle genotypes in the same media (*Şimşek et al., 2017*; *Aka Kaçar et al., 2020*).

Despite the limited number of studies in myrtle plants conducted with the Plantform temporary immersion system, there are plenty of studies conducted with solid classical cultures. *Nobre (1994)* conducted a study on micropropagation of myrtle plants with the MS media containing different concentrations of BA, NAA, and GA3 plant growth regulators. Increasing multiplication coefficients were reported with increasing BA concentrations, and multiplication coefficients varied with the culture media. *San et al. (2015)* used TDZ, BAP, and NAA growth regulators for the micro-propagation of myrtle plants and reported the most excellent multiplication coefficient as 4. *Biçen (2017)* reported a multiplication

coefficient of 6 under classical tissue culture conditions and 14 under the Plantform temporary immersion bioreactor system. Present multiplication coefficients were relatively more significant than those earlier ones. The Plantform system yielded a multiplication coefficient more significant than 21 in the Karaisalı Beyazı genotype in the present study. In brief, the Plantform temporary immersion bioreactor system positively affected myrtle plants' multiplication coefficient.

The multiplication coefficient is a significant indicator of success in tissue culture studies. Plant height is also an important criterion for success in tissue culture. The Plantform system yielded better outcomes for plant heights than solid cultures in the present study.

The rooting potentials of myrtle plants were also investigated using solid cultures, a Plantform system, and cuttings taken in two different periods. Solid cultures and the Plantform system yielded quite successful outcomes for rooting. Rooting was not encountered in scions taken in Autumn, and limited rooting was encountered in scions taken in Spring. Present findings revealed that tissue culture techniques yielded high success in rooting independently from the periods. Although myrtle plants were successfully produced from the scions, seasonally different rooting levels were reported in previous studies. The present study also observed the highly significant effects of season and genotype on Rooting percentage. To overcome the problems encountered in scion rooting, *in vitro* propagation techniques have been used, and speedy propagation has been achieved using appropriate protocols (*Canhoto, Lopes & Cruz, 1999*). Besides classical *in vitro* methods, Plantform was successfully used in the micropropagation and rooting of myrtle plants.

Micropropagation studies have been conducted using Plantform temporary immersion systems in different plant species. The first studies with the Plantform bioreactor system were performed on *Carex oshimensis* (Evergreen), *Chrysanthemum morifolium*, *Ficus carica*, and *Ribes rubrum* plants. These studies reported positive contributions of the Plantform temporary immersion system on shoot quality, multiplication coefficient, and Rooting percentage (*Lambardi et al., 2015*). In recent years, apart from myrtle plants, the Plantform system has been successfully applied to different plant species in Türkiye (*Umarusman & Aka Kaçar, 2018*; *Cengiz & Aka Kaçar, 2019*). These studies using the Plantform system yielded better outcomes than the solid cultures.

In rooting trials, 15 cm long semi-woody scions were used. To put forth the effect of the period, trials were set up with the scions taken in late Spring before flowering (May–June) and in Autumn within the fruit ripening period (October–November). Scions were taken in October 2019, and rooting trials were set up. Rooting was not encountered in all genotypes with these scions. Then, new scions were taken from the same plants in May 2020 before flowering, and rooting trials were re-set up. Rooting levels varied with the periods, and low levels were encountered. Like the present findings, *Klein, Cohen & Hebbe (2000)* indicated the significant effects of season on scion rooting. However, contrary to present findings, significantly greater rooting levels were reported for December-February than for May-August. Such differences were attributed to nutritional status, the genetic structure of the plant from which the cuttings were taken, and the ecological conditions under which this plant was grown.

Overcoming seasonal rooting problems in plant propagation requires careful consideration of various factors and the implementation of appropriate strategies. Applying growth regulators, such as auxins, can enhance root development and improve rooting success. Proper hormone concentration and timing are crucial (*Thomson & Deering, 2011*). Creating a favorable environment for rooting, including temperature, humidity, and light conditions, can significantly impact cutting success. Utilizing misting systems, humidity chambers, and controlled environments can improve rooting rates. Employing pre-rooting techniques, such as basal heat treatment, root-promoting substances, or adventitious root induction methods, can enhance rootability and overcome seasonal limitations. Choosing an appropriate rooting substrate or media provides optimal water retention, aeration, and nutrient availability. It is essential to consider the specific requirements of the plant species and adjust the substrate composition accordingly (*Cox, 2018*). It was proved that plant propagation and rooting could be achieved in any year season with proper tissue culture methods.

In addition to its application in micropropagation, TISs have been recognized for their advantages in secondary metabolite production. These advantages include high biomass production, low production cost, and the ability to generate significant quantities of metabolites compared to other standard methods (*Gueguim Kana et al., 2010*; *Misic et al., 2013*). While numerous studies have demonstrated the superiority of TISs over other liquid-phase bioreactors and Solid-State Cultivation (SSC) methods, this superiority is contingent upon careful consideration of various factors. These factors include the selection of an appropriate culture medium, optimal composition and amount of inlet air, lighting conditions within the bioreactor system, effective sterilization protocols, immersion frequency, and proper inoculation density. Consequently, improper selection or adjustment of any of these parameters may yield results contrary to expectations (*Mirzabe et al., 2022*). Secondary metabolite production in myrtle holds significant importance due to the rich bioactive compounds found in this plant species. Myrtle contains secondary metabolites such as phenolic compounds, essential oils, flavonoids, and terpenes. These bioactive compounds contribute to myrtle's medicinal, aromatic, and culinary properties. The results obtained from this study, along with the developed methods and their further improvements, have revealed a significant perspective for producing and commercializing secondary metabolites in myrtle using the Plantform system.

## CONCLUSIONS

There is an increasing demand for myrtle plants since they are used for various purposes. It is significant to achieve healthy, speedy, and economic propagation of myrtle genotypes independently from the season. Present findings proved that the Plantform system, a new-generation tissue culture system, was quite successful in the micropropagation and rooting of myrtle genotypes. Since agar is not used in these systems, a significant cost item is eliminated in tissue culture laboratories. Plantform culture plates are large and may contain more than 100 plants, and such a case brings about significant advantages in terms of cost and labor. Based on the current study's findings, we successfully implemented

*in vitro* micropropagation and rooting protocols for diverse wild myrtle genotypes, allowing for propagation independent of seasonal constraints. This approach eliminates the need for specific seasons or environmental conditions, providing a flexible and efficient method for year-round plant propagation. *In vitro* microropagation techniques involved initiating cultures from explants, followed by their subsequent multiplication through the proliferation of shoots in a controlled environment. The rooting phase encompassed the induction of adventitious roots on the microshoots, leading to the development of robust root systems. By carefully optimizing the culture conditions, nutrient media composition, and growth regulator treatments, we achieved consistent and reliable results in propagating various wild myrtle genotypes. The utilization of *in vitro* microropagation and rooting procedures in this study offers several advantages over traditional propagation methods. Firstly, it enables the rapid multiplication of plant material, producing many identical and healthy plantlets. Secondly, the technique allows for the propagation of genotypes that may exhibit poor rooting abilities through conventional cutting methods. By employing these effective *in vitro* microropagation and rooting procedures, we have established a valuable tool for the conservation, breeding, and commercial production of different wild myrtle genotypes. The flexibility and independence from seasonal limitations make this approach highly advantageous in ensuring a constant and uninterrupted supply of healthy plant material. Further studies can explore optimizing specific factors, such as culture media compositions, growth regulators, and environmental conditions, to refine and enhance the efficiency of *in vitro* microropagation and rooting protocols for different plant species. This will contribute to advancing sustainable and efficient plant propagation techniques, facilitating the preservation and utilization of genetic resources for conservation and agricultural purposes.

## ACKNOWLEDGEMENTS

The authors would like to thank Eugene Steele, professional English Editor of Erciyes University for the English language editing of the manuscript.

### Funding

This research was funded by TÜBİTAK (The Scientific and Technological Research Council of Türkiye), grant number 119O073. The funders had no role in study design, data collection and analysis, decision to publish, or preparation of the manuscript.

### Grant Disclosures

The following grant information was disclosed by the authors:
TÜBİTAK (The Scientific and Technological Research Council of Türkiye): 119O073.

### Competing Interests

The authors declare no conflict of interest.

## Author Contributions

- Özhan Şimşek conceived and designed the experiments, performed the experiments, analyzed the data, prepared figures and/or tables, authored or reviewed drafts of the article, and approved the final draft.
- Dicle Dönmez performed the experiments, prepared figures and/or tables, and approved the final draft.
- Mehmet Ali Sarıdaş performed the experiments, prepared figures and/or tables, and approved the final draft.
- Emine Acar performed the experiments, prepared figures and/or tables, and approved the final draft.
- Yıldız Aka Kaçar performed the experiments, prepared figures and/or tables, and approved the final draft.
- Sevgi Paydaş Kargı performed the experiments, prepared figures and/or tables, and approved the final draft.
- Tolga İzgü analyzed the data, authored or reviewed drafts of the article, and approved the final draft.

## Field Study Permissions

The following information was supplied relating to field study approvals (*i.e.*, approving body and any reference numbers):

The authors complied with the IUCN Policy Statement on Research Involving Species at Risk of Extinction and the Convention on the Trade in Endangered Species of Wild Fauna and Flora for collecting the plant material. The plant material was collected with the official permission numbered 43368836-335.01-E.1189927 obtained from the Ministry of Agriculture and Forestry, the Republic of Türkiye.

## Data Availability

The raw data are available in the Supplemental File.

## Supplemental Information

Supplemental information for this article can be found online at http://dx.doi.org/10.7717/peerj.16061#supplemental-information.

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
