# Peer review of "In vitro and ex vitro propagation of Turkish myrtles through conventional and plantform bioreactor systems"

_PeerJ, doi:10.7717/peerj.16061_

## Round 0.1 · original submission · Major Revisions

Dear author please revise the article and resolve the queries for each reviewer and upload response to reviewers comments file during revision.

Reviewer 1 ·

Basic reporting

The article is unambiguously written with the correct use of technical terms. The language used in the manuscript is according to Peer J standards. Some errors in manuscripts that should be corrected
For example:
Line No. 26: “and propagation and rooting” repeated in a sentence.
Line No. 38: “having” may be appropriate word in place of “and have”
Line No. 108: Write “Shoot tips were” in place of “shoot tip was”
Use only “Rooting” in place of “Cutting rooting” in entire manuscript
In result section authors have used Table and Figures title as a heading that needs to be corrected by writing tables and figures number after particular paragraphs.
Use “Rooting percentage/Percent rooting” in place of “Rooting ratio” in entire manuscript
Tables and figures provided are appropriate. However, some more photographs of Plantform bioreactor system should be included. Literature collected for research was adequate and the way of data presentation is appropriate.

Experimental design

The investigation on “In vitro and ex vitro propagation of Turkish myrtles through conventional and plantform bioreactor systems” have novelty as authors have demonstrated usefulness of Plantform bioreactor system over conventional system in a economically very important crop, Turkish myrtles. The information provided in the manuscript is sufficient for the audience to understand the usefulness of the findings.

Validity of the findings

Factorial CRD design could be appropriate design for the present investigation. However, authors have used LSD designed and data are presented in proper format, therefore we can draw proper conclusion. Enough tables of statistically analyzed data provided to explain the findings.

Additional comments

The manuscript on “In vitro and ex vitro propagation of Turkish myrtles through conventional and plantform bioreactor systems” fulfill all the requirements for the publication in Peer J. Hence, the manuscript can be accepted for publication after minor modifications as suggested.

Reviewer 2 ·

Basic reporting

Comments to the manuscript 81262v1 "In vitro and ex vitro propagation of Turkish myrtles through conventional and plantform bioreactor systems".
Authors propose the report of an experiment of propagation of myrtle using four wild genotypes, two seasons of cutting sampling, and two in vitro systems with solid medium and the plantform bioreactor. Other variables are the concentrations of the proliferation and of the rooting media. The manuscript is sufficiently well written and may be acceptable for publication after some minor changes.
The general introduction reports some uncorrect information, like in the lines 22 and 32. The myrtle is a thermophylous species mainly spreading in temperate Mediterranean and subtropical regions. It is not naturally growing in tropical areas.
I suggest the Authors to improve the introduction and the discussion with an additional work of reading of the most up to date scientific literature (e.g.: https://doi.org/10.3390/su13168785).
Line 324: please change 1996 with 1999 in the citation.
Line 394: the article of Escalona et al. 1999 was not cited in the text.
Line 372: This citation is unavailable for the international readers.

Experimental design

The experiment was correctly designed and the data treatment appropriately performed.

Validity of the findings

Results are clearly presented and correctly discussed. The manuscript may represent a good contribute to the field of study. Conclusions may be reinforced by the additional reading of the existing literature.

Additional comments

I suggest a carefull editing revision of the manuscript. Particularly, in the bibliography list there are some mistakes and names confused with the surnames.
Line 409: please change Pavlov, A. & Bley, T.
Line 414: this citation is in italian and unavailable for the international readers.

Reviewer 3 ·

Basic reporting

GENERAL COMMENTS

The aim of this research is to compared four myrtle genotypes with different genetic origins and well-regional adaptation were selected, and the performance of different rooting systems (cutting rooting, solid media rooting, and rooting with the Planform bioreactor system) was compared to achieve a healthy and speedy rooting of myrtle plants. The language of this study is understandable and grammatically quite good. Therefore, the manuscript does not need to be edited in terms of language. The introduction, discussion, and suggestion parts of the manuscript are written very well. Paragraph transitions are very convenient. Descriptive statistics of the data set used can be given. But, there is a need for important regulations in the statistical analysis part of the study. In the study, there are only major points that are overlooked. Also, the references of the study were checked again and the suitability of the journal format was left to the authors.


In line with the information given above, the study needs a major revision. After corrections are made, the manuscript does need to be re-evaluated. Specific comments on the study are given in the text.

Experimental design

Although there is no problem in the experimental planning of the study, there are problems in the presentation of the statistical analysis results.

Validity of the findings

Table 2-7 in the study should be revised, as changes are required.

Additional comments

The study needs major revision. Specific comments on the study are given in the text.

Annotated reviews are not available for download in order to protect the identity of reviewers who chose to remain anonymous.

Reviewer 4 ·

Basic reporting

The authors should mention the chances of contamination while addressing the disadvantages of the liquid culture system in lines 56-57.
As the authors have introduced the topic of problems associated with rooting due to seasonal reasons, they should also mention the references related to overcoming these problems, failing which has led them to conduct the research.

Experimental design

Complete details of plantform bioreactors should be given.
Mention references related to surface sterilization procedure especially when there are 3 minutes of prolonged exposure with 70% alcohol. Also, provide the reference for the use of 20% sodium hypochlorite.
Mention the scale bar while showing the height-related photographs of the plantlets.

Validity of the findings

Novelty can be seen in the type of propagation system. However, sufficient references should be cited.

Additional comments

The proposed work should be published as a report or short communications.

---

## Round 0.2 · accepted · Accept

Your revised article is accepted.

Reviewer 2 ·

Basic reporting

Comments to the manuscript 81262v1 "In vitro and ex vitro propagation of Turkish myrtles through conventional and plantform bioreactor systems".
Authors propose the report of an experiment of propagation of myrtle using four wild genotypes, two seasons of cutting sampling, and two in vitro systems with solid medium and the plantform bioreactor. Other variables are the concentrations of the proliferation and of the rooting media. The manuscript is sufficiently well written and may be acceptable for publication after some minor changes.
Authors changed the manuscript according to the reviewer's suggestions and it is now suitable for publication.

Experimental design

Experimental design
The experiment was correctly designed and the data treatment appropriately performed.

Validity of the findings

Validity of the findings
Results are clearly presented and correctly discussed. The manuscript may represent a good contribute to the field of study.

Additional comments

Additional comments
I suggest a carefull editing revision of the manuscript. Particularly, in the bibliography list.

Reviewer 4 ·

Basic reporting

No comment.

Experimental design

Mention references related to surface sterilization procedure especially when there are 3 minutes of prolonged exposure with 70% alcohol. Also, provide the reference for the use of 20% sodium hypochlorite.
Mention the scale bar while showing the height-related photographs of the plantlets.

Validity of the findings

No comment

Additional comments

No comment